# The Effect of Fiber Type and Yarn Diameter on Superhydrophobicity, Self-Cleaning Property, and Water Spray Resistance

**DOI:** 10.3390/polym13050817

**Published:** 2021-03-07

**Authors:** Ji Hyun Oh, Chung Hee Park

**Affiliations:** 1Department of Textiles, Merchandising and Fashion Design, Seoul National University, Seoul 08826, Korea; joh25@ncsu.edu; 2Department of Chemical and Biomolecular Engineering, North Carolina State University, Raleigh, NC 27695, USA

**Keywords:** superhydrophobicity, oleophobicity, water repellency, self-cleaning property, polydimethylsiloxane (PDMS), dip-coating, micro/micro hierarchical structure, staple fabric, filament fabric, diameter

## Abstract

In this study, we proved that micro/micro hierarchical structures are enough to achieve a superhydrophobic surface using polydimethylsiloxane (PDMS) dip-coating. Furthermore, the effect of fiber type and yarn diameter on superhydrophobicity and water spray resistance was investigated. Polyester fabrics with two types of fibers (staple fabric and filament) and three types of yarn diameters (177D, 314D, and 475D) were used. The changes in the surface properties and chemical composition were investigated. Static contact angles and shedding angles were measured for superhydrophobicity, and the self-cleaning test was conducted. Water spray repellency was also tested, as well as the water vapor transmission rate and air permeability. The PDMS-coated staple fabric showed better superhydrophobicity and oleophobicity than the PDMS-coated filament fabric, while the filament fabric showed good self-cleaning property and higher water spray repellency level. When the yarn diameter increased, the fabrics needed higher PDMS concentrations and longer coating durations for uniform coating. The water vapor transmission rate and air permeability did not change significantly after coating. Therefore, the superhydrophobic micro/micro hierarchical fabrics produced using the simple method of this study are more practical and have great potential for mass production than other superhydrophobic textiles prepared using the chemical methods.

## 1. Introduction

Superhydrophobicity has been extensively studied not only for biomedical products but also for clothing textiles because superhydrophobic surfaces can make the particular cells attract or repel water droplets easily on their surfaces [1]. Surface roughness and low surface energy are key factors to describe superhydrophobicity and self-cleaning property [2]. Superhydrophobicity represents the water repellency on the surface and when the surface has a static contact angle over 150° and a shedding angle less than 10°, the surface is regarded as having superhydrophobicity. On the other hand, self-cleaning property is the ability to remove soils from the surface by rolling water droplets [3] and this phenomenon is usually exhibited on the superhydrophobic surface. Based on lots of previous studies [4,5,6,7], when the soils are completely removed by dropping a water droplet and the dropped water goes down over 2 cm, we define that the surface has self-cleaning property. Many techniques have been developed to render surface roughness [8,9,10] and hydrophobization. To date, the micro/nano hierarchical structures has been fabricated on the surfaces via plasma etching [4,5,7,11,12,13] or chemical etching [6,14], and plasma deposition [5,15,16,17], chemical vapor deposition [18], or wet coating [1] has been used to lower the surface energy. Some studies investigated the effect of the number of monofilament [19] and yarn types (filament and Drawn Textured Yarn, DTY) [20] on superhydrophobicity. Micro/nano hierarchical roughness is important to create superhydrophobicity and self-cleaning property [21,22], but nano roughness created by these methods can be easily damaged during use, leading to a decrease in superhydrophobicity [23,24]. Thus, some superhydrophobic surfaces are made without the nano structures [25,26], because micro/micro hierarchical structures without nano roughness have high mechanical robustness and potential to be used in mass production [27,28]. Furthermore, many studies have reported that fiber type and yarn diameter of micro structures are related to the physical and mechanical properties of fabrics such as tensile strength, tearing strength, abrasion resistance, interfacial property, and flexibility [29,30,31,32,33,34]. From these results, we can know that microstructures have a stronger relationship with the mechanical properties of the fabrics, which may have a large impact on their superhydrophobicity. Therefore, the effect of microstructures should be considered in the study of superhydrophobicity. Moreover, even if textiles have superhydrophobicity, their surface wettability values vary depending on the structures of fabric, the conditions of the size and falling height of water droplets and the amount of water [35,36], and the coating performance can decay upon impact with larger droplets associated with rain (1–3 mm), which has much higher kinetic energy [37]. Thus, before using superhydrophobic textiles, a water spray test should be conducted. Therefore, in this study, we tried to reveal that nano roughness is not necessary to display superhydrophobicity and elucidate the effect of surface structures on superhydrophobicity and water spray repellency and elaborate the relationship between superhydrophobicity and water spray repellency depending on micro or micro/micro hierarchical structures. For that, five specimens with different microstructures were used by changing the fiber types, and yarn diameters. The optimum conditions were determined, and the effect of the fiber types and yarn diameters on superhydrophobicity and water spray repellency was investigated. The tested fabrics were prepared by dip-coating in PDMS solutions at three volume ratios (1, 4, and 8 *v*/*v*%) and three dip-coating times (1, 4, and 7 min). The changes in surface morphology and chemical composition were analyzed, and the add-on ratio and thickness changes were measured. Static contact angle, shedding angle, self-cleaning property with silicon carbides, and water spray repellency against silicon carbides and fine dusts were evaluated. The findings suggest that micro/micro hierarchical structure is sufficient to achieve superhydrophobic surface and the developed superhydrophobic and water spray resistant polyester fabrics have great potential for diverse textile applications due to their improved water-repellent properties.

## 2. Experiments

### 2.1. Materials

All specimens are composed of polyethylene terephthalate (PET) plain weave, but fiber type and count, twist, weave density (warp/weft), thickness, and solid volume fraction are different (Table 1). There are two types of yarns (staple and filament fibers) and three different yarn diameters (177D, 314D, and 465D). The sample codes are named according to the characteristics of the specimens (Table 1).

### 2.2. Methods

#### 2.2.1. Hydrophobization

After weighing the fabrics, the amount of tetrahydrofuran (THF, Siheung, Gyeonggi, Daejung Chemical Co., Korea) was calculated based on the liquor ratio of 1:100. Prepolymer Sylgard-184 polydimethylsiloxane (PDMS, Sylgard^®^ 184, Dow Corning Corporation, Midland, MI, USA) and its curing agent at a ratio of 10:1 were poured into the beaker filled with THF to make the concentration 1, 4, and 8 *v*/*v*%. The PDMS solutions were then sonicated for 10 min. Next, the fabrics were immersed into the PDMS solutions for 1, 4, and 7 min and cured in oven at 100 °C for 1 h. Sample codes are presented in Table 2.

#### 2.2.2. Characterization

##### Surface Morphology and Chemical Composition Change

The surfaces of the untreated and hydrophobized samples were examined using a field emission scanning electron microscope (FE-SEM, SIGMA, Carl Zeiss, Jena, Germany). To avoid the surface charging by the accumulation of electrons on the surface of non-conductive samples, Pt coating was performed using a sputter coater (EM ACE200, Leica, Austria) for 120 s at 30 mA. To analyze the changes in the chemical composition and chemical bonding condition of the samples before and after coating, X-ray photoelectron spectroscopy (XPS, AXIS-his, Kratos Inc., San Diego, CA, USA) was performed at 18 mA and 12 kV over 10 nm distance from the surfaces.

##### Add-on Ratio and Thickness Change

After conditioning at 20 °C and 65% relative humidity (RH) for 24 h, the add-on ratios of differently treated samples were measured using an analytical balance (PAG214C; OHAUS Corporation, Parsippany, United States) and calculated via Equation (1). The values averaged for five times were recorded. Thickness change before and after hydrophobization was also calculated.
(1)$$\mathrm{add}-\mathrm{on}\;\mathrm{ratio}\;\left(\%\right)=\frac{\left(Wa-Wb\right)}{Wb}*100,$$

*Wa*: weight of the dry specimen after treating.

*Wb*: weight of the dry specimen before treating.

##### Superhydrophobicity, Self-Cleaning Property, and Water Spraying Test

The surface wettability was determined from the static contact angle and the shedding angle. The static contact angle was measured using a Theta Lite optical tensiometer (KSV Instruments, Helsinki, Finland). The sample attached on a slide glass with Scotch tape at both ends was positioned on the sample stage of the equipment. Then, 3.0 ± 0.5 μL water droplets were dropped on the sample, and the contact angles of the water droplets were measured. The minimum angle when the water droplet of 12.5 μL dropped from a height of 1 cm above the sample rolled down more than 2 cm was determined as the shedding angle [38]. The measurement of the static contact angle and shedding angle was repeated five times and the results were averaged. Water and isopropyl alcohol (Daejung Chemical Co., Gyeonggi, Korea) were mixed at the volume ratio of 95:5, 90:10, 80:20, and 40:60 to prepare liquids with the surface tension of 50, 42, 33, and 24 dyne/cm, respectively. The self-cleaning property was modified to improve the accuracy compared to the previous studies [39]. Therefore, 0.08 g of silicon carbide (Yakuri Pure Chemicals Co., Kyoto, Japan) that served as dirt was spread on the samples placed for the angle measurement. Then, 12.5 μL water droplets were dropped on the samples using the same method as for the shedding angle. The rolled lengths of the water droplets were measured. When the dirt was clearly removed by the water droplet, which was rolled further than 2 cm at the angle below 10°, the sample was considered to have self-cleaning. Water spraying test was conducted using the standard test method AATCC 22-2010, and self-cleaning properties using silicon carbide particles and fine dusts (ISO 12103-1, Powder Technology Inc., Minnesota, USA) were also tested following the water spraying test.

##### Water Vapor Transmission Rate (WVTR) and Air Permeability

Before and after hydrophobization, the WVTR and air permeability were measured in accordance with ASTM E96-80 and ASTM D737-75. For WVTR, three different fabrics were placed on the preheated at 40 °C ± 2 °C and filled with 33 g of anhydrous calcium chloride (YAKURI) cups. The cups were placed in a conditioned chamber maintained at 20 °C ± 2 °C and RH 65% ± 5%. After 1 h, the cups were weighted and then 1 h later, the weight change was recorded again. The average WVTR was calculated. As for air permeability, a specimen with the dimension of 12 cm × 12 cm was placed in an air permeability tester (FX 3300, TEXTEST, Schwerzenbach, Switzerland) and forced under 125 Pa pressure. The air permeability was measured three times, and the average values were reported in cfm (cubic feet per minute).

## 3. Results

### 3.1. Surface Properties by the Fiber Type

Staple and filament fabrics with similar yarn diameters, cover factors, cloth cover factors, thickness, and solid volume fraction, were selected to investigate the effect of fiber type on surface wettability. By comparing them, the effect of micro and micro/micro hierarchical structures were investigated on surface wettability.

#### 3.1.1. Changes in Surface Morphology, Add-on Ratio, and Thickness

As shown in Figure 1a and Appendix A, the untreated staple fabric had extra microstructures stemmed from the slippage of fiber tips where they cannot be gripped by twisting yarns during weaving [40]. Contrarily, the untreated filament fabric had a continuously smooth surface [41] (Figure 1e and Appendix A). After 1 *v*/*v*% PDMS coating for 1 min, staple fibers were stacked together (Figure 1d) because the protruded fibers create gaps between them that can be penetrated by the PDMS solutions. By contrast, the surface of the filament fabric did not change after PDMS coating and the long and smooth filament fibers were kept separately (Figure 1e–h) because filament fibers do not have the protruded region to hold the solutions. Therefore, the staple fabric had a higher rate of increase in the thickness but a lower rate of increase in the add-on ratio than the filament fabric (Figure 2) because the coating occurs by occupying interfilament spaces with solutions in the form of capillaries. Therefore, solutions can spontaneously wick better into the filament fabric, which has continuous pores and longer capillary length than the staple fabric [42,43]. Thus, although the staple fabric had a large thickness change than the filament fabric as short yarns showed aggregation on the surface by a low capillary force, the coating solutions covered the smooth fiber type of the filament fabric more uniformly owing to its high wicking property, leading to a greater increase in the add-on ratio.

#### 3.1.2. Changes in Chemical Composition

The changes in the surface chemical composition were measured via energy-dispersive X-ray spectroscopy (EDS) [44] and X-ray photoelectron spectroscopy (XPS) to obtain the surface properties related to superhydrophobicity. Independent of the fiber type, the untreated fabrics consisted of carbon and oxygen [23] (Figure 3a,c and Appendix A). However, after P1T1 PDSM coating, both fabrics contained silicone (Si) on the surface [45], but the filament fabric had higher Si content (1.19% in Figure 3 and 21.3% in Appendix A) than the staple fabric (0.47% in Figure 3 and 11.8% in Appendix A). This result suggests that long and smooth filament fibers with continuous interfiber capillaries can absorb the PDMS solution better than the short and protruded staple fibers [42,43], which agrees with the previous result of the add-on ratio and the water spray test.

#### 3.1.3. Superhydrophobicity

The photographs of water droplets on the untreated and P1T1 PDMS-coated staple and filament fabrics are shown in Figure 4. The untreated staple fabric immediately absorbed a water droplet owing to its own micro/micro hierarchical structure, whereas the untreated filament fabric with a smooth surface took 15 s to absorb the water droplet completely. This observation can be explained by the Wenzel equation [46]; for a hydrophilic (hydrophobic) surface, roughness decreases (increases) the static contact angle and thus the hydrophobic surface exhibits higher water repellency.

After P1T1 PDMS coating, water droplets kept their spherical shape on the treated polyester fabrics irrespective of the yarn type. Based on the PDMS concentrations and coating durations, surface hydrophobicity of the fabrics was measured using static contact angles and shedding angles. As shown in Figure 5a, static contact angles of the filament fabric decreased with the increase in the PDMS concentration and coating time, caused by covering the surface with the solution uniformly. However, the staple fabric maintained the increased static contact angles over 160° because short yarns of the staple fabrics help them have micro/micro hierarchical structures and sufficient roughness to maintain high static contact angles (Appendix A). All the shedding angles of the filament fabrics were over 10° regardless of the PDMS concentration and coating duration, while a shedding angle below 10° (9.2° ± 0.3°) was achieved on the staple fabrics under the condition of 1 *v*/*v*% and 1 min PDMS coating (Figure 5b). In the staple fabrics, the increase in the shedding angles was higher than the decrease in the static contact angles according to the PDMS concentrations and coating durations. This is because the surface roughness decreased by covering the surface with the solution at larger PDMS concentrations and longer coating durations. Therefore, the shedding angle, which is more affected by the surface roughness showed a larger increase than the decrease in the static contact angle. However, the staple fabric had a static contact angle over 150° and a shedding angle less than 10° in P1T1 PDMS coating; thus, the micro/micro hierarchical structure was sufficient to meet Cassie-Baxter model and accomplish superhydrophobicity. Because P1T1 PDMS-coated staple fabric had enough air gaps between micro/micro hierarchical structures and pressure to push a water droplet [47].

Furthermore, static contact angles were measured with the liquids of decreasing surface tensions. As a result, P1T1 PDMS-coated staple fabric had higher static contact angles than the filament fabric at the same condition, showing oleophobicity by keeping over 150° until the surface tension reaches 42 dyne/cm [48]. However, the P1T1 PDMS-coated filament fabric lost its oleophobicity at the surface tension of 55 dyne/cm, showing the static contact angle of 135.5° ± 3.7°. Based on this result, the existence of the micro/micro hierarchical structure is crucial to maintain superhydrophobicity and oleophobicity (Appendix A).

#### 3.1.4. Self-Cleaning Property

Removing the soils on the surface by rolling water droplets is important for superhydrophobic textiles [49] to be widely applied in daily life because it shortens the number of washing cycles, reduces energy consumption, and enhances the care convenience and environmental friendliness [1,4,5,6,7,11,14]. In this study, the rolling-down distances of water droplets were measured on the staple and filament fabrics contaminated without and with silicon carbides (Appendix A). At 10°, the P1T1 PDMS-coated staple fabric repelled a water droplet by rolling it over 3 cm (Figure 6a), but silicon carbides were not eliminated by dropping a water droplet because it keeps its spherical shape at the right dropped position (Figure 6b). By increasing the tilted angle of the specimen to 15°, water droplets rolled down over 3 cm and silicone carbides were removed, but they still remained on the surface because they penetrated and fixed into the space between the short fibers (Figure 6c). Therefore, when the fiber type is staple, the surface has poor self-cleaning properties due to higher trapping of particles onto surface of the hairy micro/micro hierarchical structures made of short fibers. On the P1T1 PDMS-coated filament fabric, a water droplet rolled down for about 0.45 cm (Figure 6d and Appendix A), whereas a water droplet rolled down for about 0.77 cm when the filament fabric was contaminated with silicon carbides (Figure 6e and Appendix A). This result was also observed when the filament fabric was inclined to 15°. At 15°, the P1T1 PMDS-coated filament fabric made a water droplet roll down for over about 2.33 ± 0.28 cm, but the water droplet approximately rolled down for over 3 cm after silicon carbides were spread on the surface. (Figure 6f,g and Appendix A). These might be caused by the increase in the weight and gravity force of the water droplets after silicon carbide absorption and the smooth surface of the filament fabric. Compared to the staple fabric, which has a micro/micro hierarchical structure, the filament fabric exhibited better self-cleaning property due to its smooth surface. This is because protruded fibers anchor the resin in the interfacial region, leading to a higher interlaminar adhesion and greater interfacial shear strength than smooth fibers [32]. The staple fabric has higher affinity for particles than the filament fabric, and particles are fixed between the protruded fibers and thus the staple fabric has worse self-cleaning properties. The filament fabric has a smooth surface and thus is more favorable than the staple fabric. Regardless of the yarn type, none of the untreated fabrics could remove silicon carbides because they absorb the water droplets instantly at 10° and 15° (Appendix A).

Self-cleaning properties were measured using silicon carbides and fine dusts with the spray testing equipment. Untreated staple fabric could not remove silicon carbides and fine dusts when 250 mL water was poured over it, making residual particles on the surface wet because of its hydrophilicity and the micro/micro hierarchical structure (Figure 7a,c). In contrast, P1T1 PDMS-coated staple fabric eliminated silicon carbides and fine dusts, but few stacked droplets and contaminants stayed on the surface because its extra microstructure formed with protruded staple fibers blocked the contaminants from rolling down and washing (Figure 7b,d). Untreated filament fabric removed silicon carbides upon pouring water due to its smooth surface, but the particles stayed at the bottom area of the specimens and became entirely wet owing to their hydrophilicity (Figure 7e). However, after the P1T1 PDMS coating, the filament fabric removed silicon carbides perfectly and remained dry (Figure 7f). Untreated filament fabric could not eliminate fine dusts and particles smaller than silicon carbides and thus became wet (Figure 7g). The P1T1 PDMS-coated filament fabric removed fine dusts perfectly because of its low surface energy and continuous smooth surface (Figure 7h). Therefore, after P1T1 PDMS coating, in all specimens, self-cleaning property combined with the water spray test was improved because the adhesion between contaminants and fabrics decreased by covering the fabrics with a low-surface-energy material, PDMS. Moreover, filament fabrics show better self-cleaning properties than the staple fabrics owing to their surface structures. These results suggest that the self-cleaning property varies with the surface roughness arising from the fiber type, particle size, and surface energy.

#### 3.1.5. Water Spray Test

The other method to test the water repellency of the fabric is the water spray test [50] because the drop impact from a high level can distinguish the wettability related differences that the classic static contact angle and shedding angle cannot [51,52]. To date, a single droplet has been typically used for impact durability [53], but the prolonged exposure to the high-speed and large-quantity droplets should also be considered for practical use [37]. Thus, in this study, for the water spray test, 250 mL water dyed with a hydrophilic blue ink was used. Regardless of the fiber type, all untreated fabrics became wet entirely and left blue ink marks on the surface because they originally have hydrophilicity and immediately absorb water droplets (Figure 8a,c). However, the dyed marks were more apparent on the untreated staple fabric than untreated filament fabric because staple fibers have higher adhesion between fibers than filament fibers due to the protruded fibers [54]. Therefore, the difference in the amount of remained stains is caused by the micro/micro hierarchical staple fabric that can easily trap and retain more dyes between fibers than the smooth filament fabric. After P1T1 PDMS coating, the staple fabric presented ISO3 water spray resistance and the filament fabric repelled the water droplets almost completely on the surface and showed ISO4 water spray resistance (Figure 8b,d). The P1T1 PDMS-coated staple fabric and filament fabric showed superhydrophobicity and hydrophobicity, and the filament fabric showed better water spray resistance than the staple fabric. This can be explained by the uniform coverage of the coating and maximum pressure, *p*^max^, meaning the highest pressure that the surface can endure. The staple and the filament fabric differ by the uniformity of the coating due to the difference in the capillary force. The staple fabric is not coated evenly due to the short fibers making the capillary path cut, preventing the coating solution penetrating the inside of the yarns. However, the continuous filament fabric was uniformly covered with the solution due to the high capillary force, consistent with the results of the add-on, EDS and XPS analysis (Figure 2 and Figure 3 and Appendix A). 

Moreover, the drop impact consists of four phases: approaching, spreading, receding (recoiling or bouncing), and equilibrium motion, which is sensitive to the impact and surfaces conditions [37,51,52]. The fabric with *p*^max^ lower than or equal to the water pressure goes through irreversible transition and water droplets are pinned on the surface, but the fabric with *p*^max^ higher than the applied pressure can repel and bounce the water droplets, generating energy dissipation owing to its lower surface friction [53]. Thus, the uniformly coated filament fabric with *p*^max^ higher than the irregularly coated staple fabric can endure against the applied external pressure and therefore show better water spray resistance. These results suggest that shedding angles and static contact angles are affected by surface roughness, while water spray resistance is influenced by the uniformity of the coating and *p*^max^.

#### 3.1.6. WVTR and Air Permeability

One of drawbacks of the coating process is the decrease in the WVTR and air permeability because the pores of the fabric are covered with the solution [55,56]. The WVTR and air permeability after PDSM dip-coating slightly decreased regardless of the fiber type because both sides of the fabrics were coated and the pores were filled with the solutions, blocking the transport of water vapor and air [57,58,59]. However, the reduction of the WVTR and air permeability was not significant (Figure 9). Therefore, the P1T1 PDMS-coated staple fabric had not only superhydrophobicity and oleophobicity but also satisfactory WVTR and air permeability values.

### 3.2. Surface Properties According the Yarn Diameter

The effect of the yarn diameter on superhydrophobicity was determined by choosing three specimens (177D, 314D, and 475D) because these staple fabrics have different yarn diameters with similar cover factors and cloth cover factors (Table 1).

#### 3.2.1. Changes in Surface Morphology, Add-on Ratio, and Thickness

The changes in the surface morphology before and after PDMS coating are shown in Figure 10. The diameter difference was obvious among 177D, 314D, and 475D in lower magnification (Figure 10a,e,i). In the fabric with the thinnest yarns (177D), the higher add-on ratio and thickness changes were observed (Figure 11) because the PDMS solution penetrated the yarns easily. Correspondingly, 177D had an increased thickness after PDMS coating, whereas 314D and 475D showed a decreased thickness because the thicker yarns require more coating time or higher PDMS concentrations to cover the surface than the thinner yarns. Therefore, at the same coating time and PDMS concentration, the thicker yarn could not be fully coated with PDMS, which penetrates the protruded short yarns on the surface. Therefore, PDMS-coated 314D and 475D showed a smaller increase in the add-on ratios and a negative thickness change compared to 177D (Figure 11).

#### 3.2.2. Change in Chemical Composition

The change in the surface chemical composition is shown in Figure 12 and Appendix A. All the untreated 177D, 314D, and 475D had only carbon and oxygen [60] and P1T4 PDMS-coated 177D, 314D, and 475D had carbon, oxygen, and silicone on the surface [61]. However, the amount of silicone detected on the surface of the specimens was slightly different. With the thickest yarn, 475D had the lowest silicone concentrations (1.58% in Figure 12 and 14.1% in Appendix A) on the surface followed by 314D (1.96% in Figure 12 and 19.9% in Appendix A) and 177D (2.30% in Figure 12 and 22.9% in Appendix A); thus, the fabric with the thicker yarn was coated less on the surface than the fabric with the thinner yarn under the same conditions; i.e., PDMS concentration and coating duration. This result agrees with the results of the add-on ratio, thickness, and superhydrophobic property in the later section.

#### 3.2.3. Superhydrophobicity

The static contact angles and shedding angles were measured according to PDMS concentrations (1, 4, and 8 *v*/*v*%) and coating durations (1, 4, and 7 min) (Table 3 and Table 4). When PDMS concentrations and coating durations increased, 314D and 415D had high static contact angles regardless of the treatment condition, because more Si components were coated on the surface (Appendix A). However, depending on the PDMS concentrations or coating durations, 177D showed a decrease and increase in the static contact angles and the shedding angles, respectively (Table 3 and Table 4). Because 177D has relatively smaller size of yarn than others, its surface is entirely covered with the PDMS solution, resulting in a decreased surface roughness. Under the 1 *v*/*v*% and 1 min PDMS condition, 177D showed a static contact angle of 168.1° ± 3.0°, but the static contact angles of 314D and 475D were 0.0° ± 0.0°. The static contact angles of 314D and 475D gradually increased with the PDMS concentrations in all coating durations. This result indicates that the relatively thicker yarns need more time to coat their entire surface than the thinner yarns.

Shedding angles showed similar results to those of the static contact angles. 177D exhibited shedding angles of 8.8° ± 0.8° and 8.5° ± 0.0° at the PDMS concentration of 1 *v*/*v*% and coating duration of 1 and 4 min, respectively. However, after 7 min, the shedding angle increased. The inclination of the shedding angle was also the same under the conditions of 4 and 8 *v*/*v*% concentrations as the coating durations were longer. Thus, when the PDMS solutions excessively coated the surface of the fabric, the micro/micro hierarchical structures decreased the roughness owing to the adherence between the short yarns. Consequently, higher concentrations and longer coating durations are not favorable to fabricate superhydrophobic surfaces when the yarn is relatively thin. In contrast, 314D and 475D whose yarns are relatively thick showed decreased shedding angles according to the PDMS concentrations and coating durations because their surface was coated more evenly. Under the condition of 1 *v*/*v*% and 4 min (P1T4), 177D presented the highest static contact angle of >150° and the lowest shedding angle of <10° compared to 314D and 475D, which suggests that smaller sized yarns have more hydrophobic properties [62]. Therefore, the 1 *v*/*v*% and 4 min PDMS condition was selected as the optimum condition.

At the optimum condition, static contact angles were evaluated according to the surface tensions of the droplets (Appendix A). All the specimens sustained over 150° of static contact angles until the surface tension of the liquid decreased to 33 dyne/cm, but the droplets of 24 dyne/cm were immediately absorbed. However, the P1T4 PDMS-coated 314D and 475D had comparable static contact angles, while P1T4 PDMS-coated 177D showed lower static contact angles than the others. As previously reported [41,63,64,65,66], Δ*P*, intrusion pressure, which should be overcome for the penetration can be expressed as follows [63,67].
(2)$$\Delta P=\frac{2\gamma }{R}=-l\gamma \left(\cos\theta_{a}\right)/d,$$
where *γ* is the interfacial tension, R is the radius of the meniscus, *l* is the membrane pore’s circumstance, *θ_a_* is the advancing contact angle of the liquid on the surface, and *d* is the average diameter of the pore. In Equation (2), the superhydrophobic surface with *θ_a_* > 90° has Δ*P* > 0 and liquids are blocked to penetrate. Contrarily, the hydrophilic surface with *θ_a_* < 90° and Δ*P* < 0 allows the liquids to pass through the pores by gravity. Moreover, Δ*P* is dependent on the size of the pore and consequently the surface with a larger *d* has lower Δ*P*, meaning that the liquid can easily go through the surface compared to the surface with a smaller *d* and higher Δ*P*. Therefore, the P1T4 PDMS-coated 177D exhibiting relatively larger pore size and lower Δ*P* presented lower static contact angles than the other specimens depending on the surface tension of the liquids (Figure 10 and Appendix A).

#### 3.2.4. Self-Cleaning Property

P1T4 PDMS-coated 177D without silicone carbides repelled a water droplet dropped at 10° (Figure 13a), but after silicon carbides were spread, 177D did not show self-cleaning properties by water droplet that rolled down for only 0.50 ± 0.26 cm (Figure 13b and Appendix A). This means that silicone carbides that remained in the gaps between short staple fibers prevent water droplets from going down. By increasing the tilt angle to 15° and 20°, water droplets rolled down further and left tracks but silicone carbides were not removed completely (Figure 13c,d). The same was also observed with 314D; a water droplet rolled down for over 3 cm on the surface of the P1T4 PDMS-coated 314D (Figure 13f and Appendix A), but the water droplet was almost stuck when silicone carbides were spread on the P1T4 PDMS-coated 314D (Figure 13g). At 15°, the water droplet rolled down for over 3 cm and eliminated silicon carbides partially. The P1T4 PDMS-coated 475D rolled the water droplets down for about 1.17 ± 0.15 cm at 10° (Figure 13h and Appendix A), but at 15°, water droplets rolled down for over 3 cm (Figure 13j). After spreading silicone carbides, the water droplet rolled down less than the P1T1 PDMS-coated 475D without silicon carbides (Figure 13i,k). After spreading silicon carbides, the water droplet rolled down more at 15° than at 10°, but silicon carbides were not removed entirely. These observations also support the previous result that silicon carbides that were stuck between the staple fibers have the affinity for the fabric and prevent the water droplet from rolling down. After 177D and 314D are contaminated with silicon carbides, water droplets rolled down faster with the increasing yarn diameter (Figure 13a–g) because the porosity becomes larger as the fiber diameter decreases [68], leading to an increase in the exposure to silicon carbides. All untreated 177D, 314D, and 475D absorbed the water droplets and could not eliminate silicone carbides (Appendix A).

#### 3.2.5. Water Spraying Test

The conditions, where 177D and 314D showed superhydrophobicity, were chosen for the water spraying test. In the selected setting, 475D showed hydrophobicity, not satisfying the required standard of the shedding angle. The P1T4 PDMS-coated 177D displayed the level of ISO2, while P1T4 PDMS-coated 314D showed the level of ISO4 (Figure 14a,b). The P1T4 PDMS-coated 475D exhibited the level of ISO3 (Figure 14a,c). Therefore, the thinnest yarn (177D) showed superhydrophobicity but did not have a beneficial effect on the water spray resistance. Furthermore, although 475D had lower hydrophobicity than 177D, it had better water spray resistance (Figure 14f) because 177D had larger pores than 475D as shown in the images of confocal microscope (Figure 14d,f). Consequently, water droplets released at the height of 15 cm easily went through the backside of 177D due to its high porosity [31], causing the low water spray resistance. However, 475D did not have higher water spray resistance than 314D because PDMS solutions in the selected dip-coating duration or concentrations cannot cover all the surfaces of the fabric. Thus, the specific part of 475D, the right-upper part in the picture, could not repel and absorb the water droplets (Figure 14c), indicating that thicker yarn needs a longer dip-coating time or higher PDMS concentrations to show good water spray repellency. When comparing 177D and 314D which coated uniformly on the surfaces, the dynamic behavior of water droplets can be illustrated with the capillary and dynamic pressures. Capillary pressure (*P_c_*) can be defined as follow [69].
(3)$$P_{c}\approx \frac{2\sqrt{2}\sigma \cos\theta }{S},$$
*σ* is the surface tension of the liquid, θ is the static contact angle of the surface, and *S* is the spacing of the roughness. In case of the dynamic pressure (*P_D_*), it is expressed as follow [69].
(4)$$P_{D}=0.5\rho V^{2},$$
ρ is the density of liquid and *V* is the impact velocity. Therefore, in order to rebound water droplets and not to wet the surface, the capillary pressure should be larger than the dynamic pressure. Considering the density of liquids (ρ) and the impact velocity (*V*) were same and the static contact angles (θ) of P1T4 PDMS-coated 177D and 314D were similar; 168.3° ± 2.6° and 168.0° ± 1.6°, the spacing of the roughness (*S*) will be critical. Thus, as seen in Figure 14d,e, P1T4 PDMS-coated 177D with the larger spacing is expected to have lower capillary pressure (*P_c_*) than P1T4 PDMS-coated 314D so that P1T4 PDMS-coated 177D exhibited poor water resistance than P1T4 PDMS-coated 314D. From these results, superhydrophobicity is influenced by surface roughness [70,71], while water spray resistance is affected by porosity and coating uniformity. A previous study reported that plasma-treated polyester fabrics have high repellency against oil but poor water spray resistance [72]; thus, the method used in this study is more favorable to water spray resistance than the plasma treatment. In this study, we analyzed the PDMS concentrations, dip-coating durations, and fabric specs to achieve superhydrophobicity as well as water spray resistance and showed that the P1T4 PDMS-coated 314D satisfied the static contact angle of 165.7° ± 1.6°, the shedding angle of 8.8° ± 0.3°, and the spray testing level of 4 (Figure 14b,e).

#### 3.2.6. WVTR and Air Permeability

After P1T4 PDMS coating, all the specimens had slightly decreased WVTRs and air permeabilities (Figure 15) because the PDSM solution covered the areas between pores [57,58,59]. However, the decrease was not significant and the values were within the margin of error. As a result, P1T4 PDMS-coated 314D showed not only superhydrophobicity but also water repellent properties and clothing comfort.

## 4. Conclusions

This study compared superhydrophobicity and water spray repellency based on the fiber types and yarn diameters. Regardless of the fiber types, filament fabric and staple fabric showed decreased static contact angles and increased shedding angles by increasing the PDMS coating concentrations and durations. The P1T1 PDMS-coated filament fabric did not satisfy superhydrophobicity, with the shedding angle of >10°. However, the staple fabric achieved superhydrophobicity with a static contact angle of 167.7° ± 1.1° and a shedding angle of 9.2° ± 0.3° because the micro/micro hierarchical surface made by staple fibers has a positive impact on surface roughness. Furthermore, due to the micro/micro dual structures, the P1T1 PDMS-coated staple fabric showed oleophobicity until the surface tension of 42 dyne/cm is reached. For the self-cleaning property, at the tilted angle of 10°, the P1T1 PDMS-coated staple fabric could not remove silicone carbides by water droplets due to the voids made by staple fibers. At 15°, P1T1 PDMS-coated filament fabric contaminated with silicone carbides showed satisfactory self-cleaning properties; it eliminated silicone carbides completely and rolled down the water droplet to the edge, but the uncontaminated P1T1 PDMS-coated filament fabric could not roll down a water droplet over 2 cm. When the self-cleaning test was conducted with the water spray test using silicone carbides and fine dust, P1T1 PDMS-coated staple fabric removed silicone carbides entirely but some fine dust remained; however, the filament fabric eliminated both contaminants. As for the water spray repellency, P1T1 PDMS-coated staple fabric and filament fabric showed level ISO3 and level ISO4, respectively. Therefore, the micro/micro hierarchical surface of the staple fabric is more favorable to fabricate superhydrophobic surfaces than the smooth surface of the filament fabric, but the smooth surface of the filament fabric has better water spray repellency and self-cleaning property than the staple fabric. Air permeability did not change significantly after PDMS coating. 

As the yarn diameter increases (177D, 314D, and 475D), the fabric needs higher PDMS concentrations and longer PDMS coating durations to obtain static contact angles over 150°. Under selected conditions, all the specimens presented oleophobicity up to 33 dyne/cm, residing in the micro/micro hierarchical structure. Water spray repellency showed that 314D had level ISO4 but 177D and 475D had level ISO2 and ISO3, respectively. Therefore, the P1T4 PDMS-coated 314D had not only superhydrophobicity but also good water spray repellency. 

The results suggest that superhydrophobicity can be achieved with micro/micro hierarchical structures and the mono microstructure is favorable for the self-cleaning property and water spray repellency. Furthermore, superhydrophobic and water spray repellent fabric can be achieved by modifying the micro/micro hierarchical structures only by simple PDMS coating. Thus, the developed fabrics in this study have high potential for applications in various fields such as clothing, biomaterials, and industry. For the future work, it will be interesting and meaningful if the modeling and simulation are utilized for the illustration for the phenomenon of the static contact angle. 

## Figures and Tables

**Figure 1 polymers-13-00817-f001:**
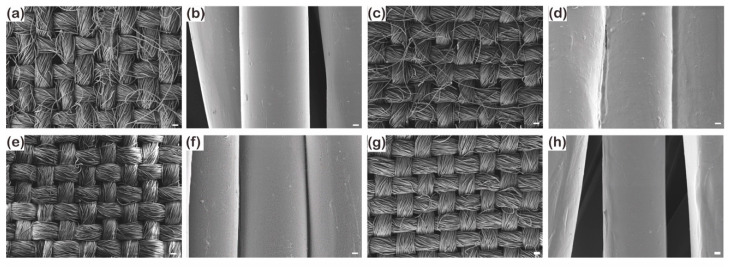
SEM images of untreated (**a**,**b**) and P1T1 PDMS-coated (**c**,**d**) staple fabrics, untreated (**e**,**f**) and P1T1 PDMS-coated (**g**,**h**) filament fabrics. (magnification: ×100 and scale bar: 200 µm and magnification: ×10,000 and scale bar: 2 µm).

**Figure 2 polymers-13-00817-f002:**
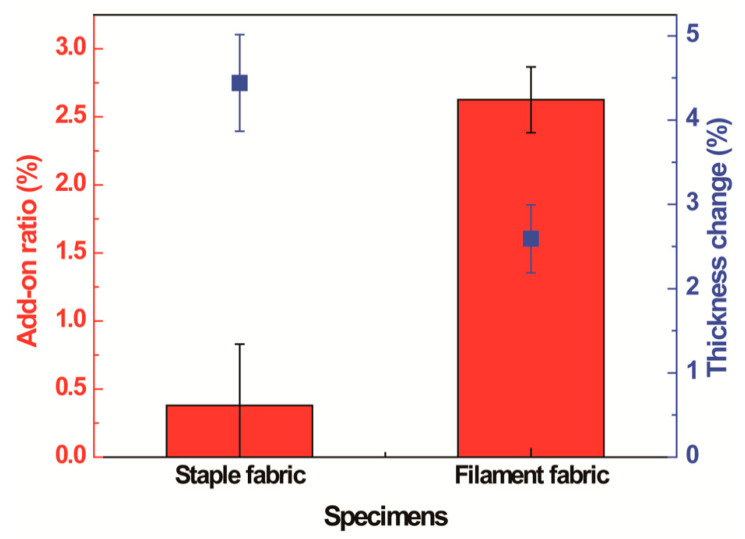
Change in the add-on ratio and thickness of staple fabric and filament fabric after 1 *v*/*v*% PDMS coating for 1 min (P1T1).

**Figure 3 polymers-13-00817-f003:**
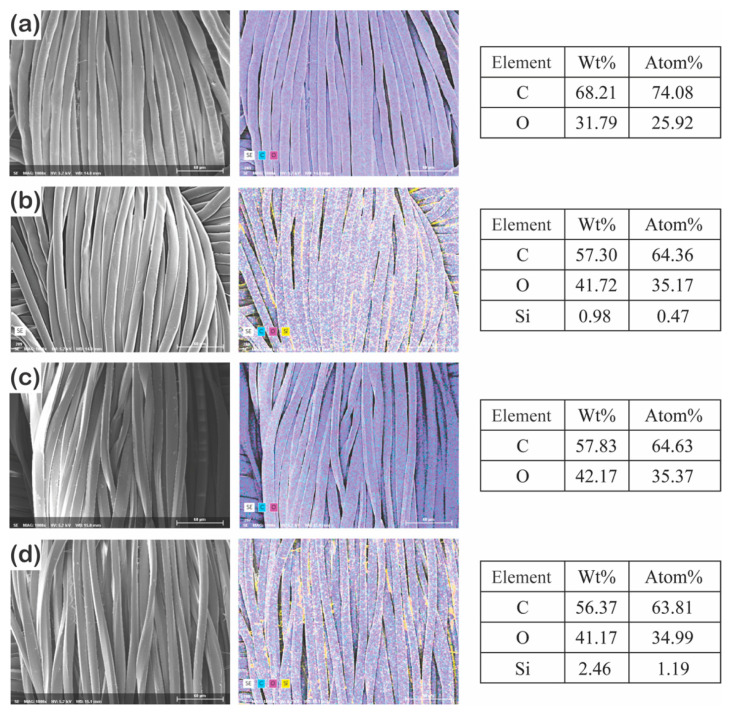
Surface chemical composition of untreated (**a**) and P1T1 PDMS-coated (**b**) staple fabrics and untreated (**c**) and P1T1 PDMS-coated (**d**) filament fabrics [different colors indicate different elements such as carbon (**c**) in light blue; oxygen (O) in magenta; and silicone (Si) in yellow]. (×1000).

**Figure 4 polymers-13-00817-f004:**
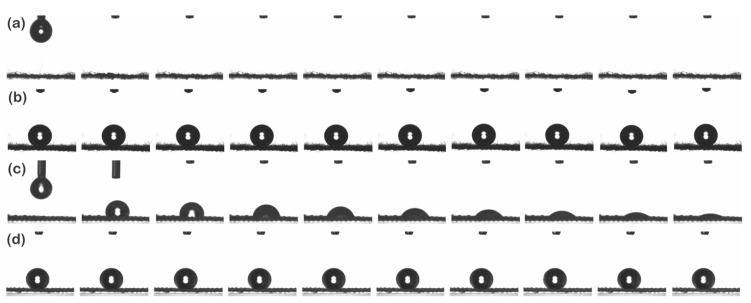
Photographs of water droplets on the surface of the untreated (**a**) and P1T1 PDMS-coated staple fabrics (**b**), untreated (**c**) and P1T1 PDMS-coated filament fabrics (**d**).

**Figure 5 polymers-13-00817-f005:**
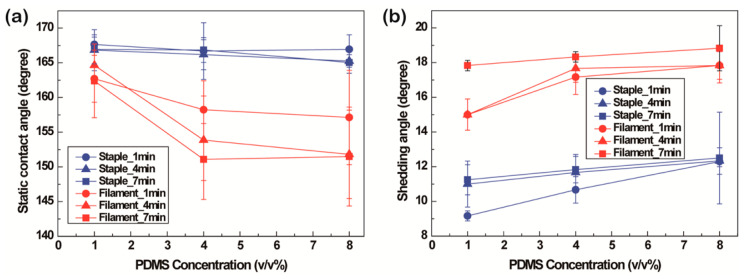
Static contact angles (**a**) and shedding angles (**b**) of staple and filament fabrics after 1, 4, and 8 *v*/*v*% PDMS coating for 1, 4, and 7 min.

**Figure 6 polymers-13-00817-f006:**
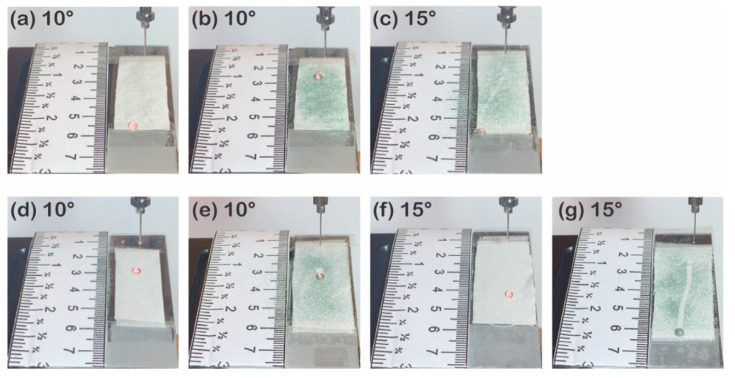
Self-cleaning tests of P1T1 PDMS-coated staple fabric without silicon carbides tilted at 10° (**a**) and with silicon carbides tilted at 10° (**b**) and 15° (**c**) and P1T1 PDMS-coated filament fabric without (**d**) and with **(e)** silicon carbides tilted at 10° and without **(f)** and with **(g)** silicon carbides tilted at 15°.

**Figure 7 polymers-13-00817-f007:**
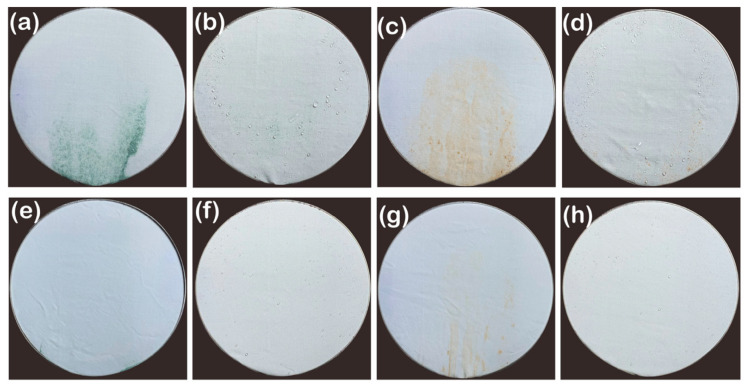
Self-cleaning tests combined with the water spray test of fabrics: untreated staple fabric (**a**) and P1T1 PDMS-coated staple fabric (**b**) contaminated with silicon carbides, untreated staple fabric (**c**) and P1T1 PDMS-coated staple fabric (**d**) contaminated with fine dusts, (**e**) untreated filament fabric (**e**), and P1T1 PDMS-coated filament fabric (**f**) contaminated with silicon carbides, untreated filament fabric (**g**), and P1T1 PDMS-coated filament fabric (**h**) contaminated with fine dusts.

**Figure 8 polymers-13-00817-f008:**
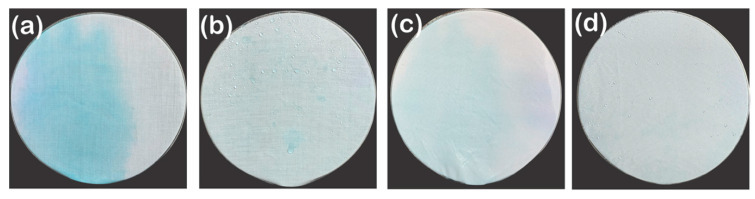
Photographs of fabrics after water spraying test: untreated (**a**) anP1T1 PDMS-coated staple fabric (**b**) and untreated (**c**) and P1T1 PDMS-coated filament fabric (**d**).

**Figure 9 polymers-13-00817-f009:**
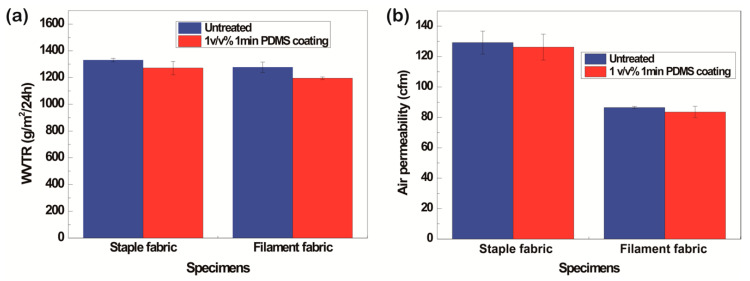
WVTR **(a)** and air permeability **(b)** of untreated and P1T1 PDMS-coated staple fabrics and filament fabrics.

**Figure 10 polymers-13-00817-f010:**
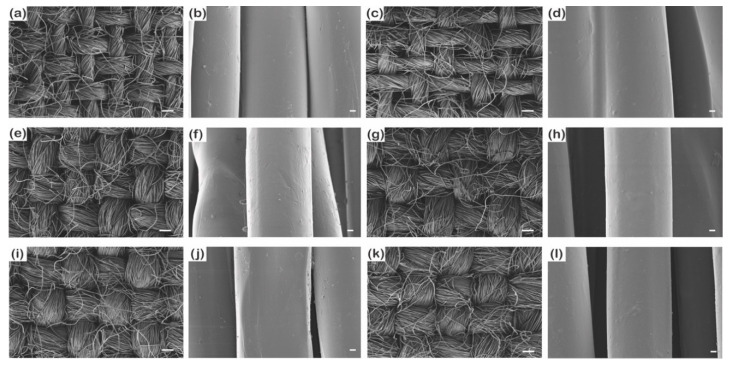
SEM images of untreated 177D (**a**,**b**), 314D (**e**,**f**) and 475D (**i**,**j**) and 1% 4 min PDMS-coated 177D (**c**,**d**), 314D (**g**,**h**), and 475D (**k**,**l**). (magnification: ×100 and scale bar: 200 µm and magnification: ×10,000 and scale bar: 2 µm).

**Figure 11 polymers-13-00817-f011:**
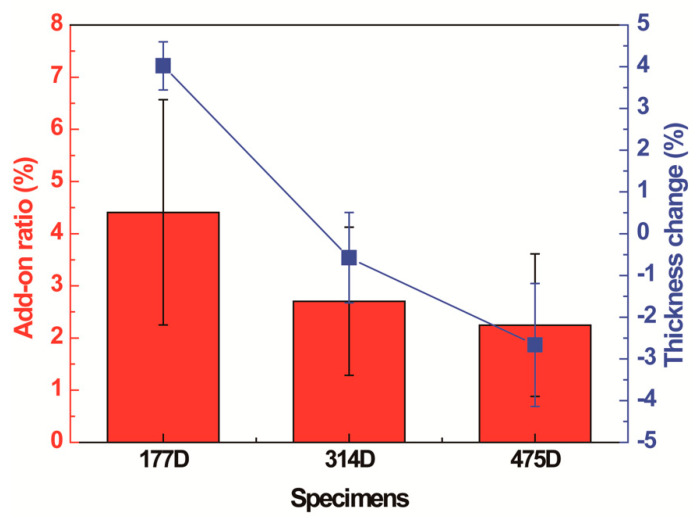
Changes in the add-on ratio and thickness of 177D, 314D, and 475D after 1 *v*/*v*% PDMS coating for 4 min.

**Figure 12 polymers-13-00817-f012:**
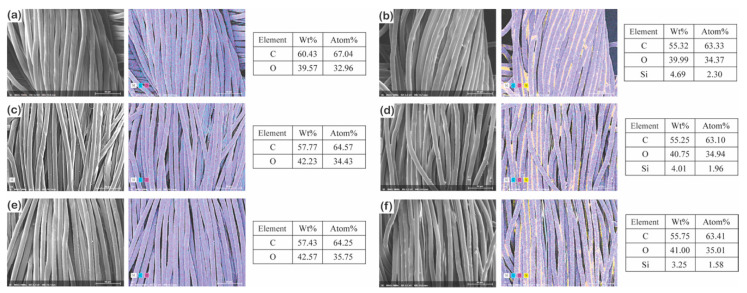
Surface chemical composition of untreated (**a**) and P1T4 PDMS-coated 177D (**b**), untreated (**c**), and P1T4 PDMS-coated 314D (**d**) and untreated (**e**) and P1T4 PDMS-coated 475D (**f**). Different colors indicate different elements such as carbon (C) in light blue; oxygen (O) in magenta; and silicone (Si) in yellow (×1000).

**Figure 13 polymers-13-00817-f013:**
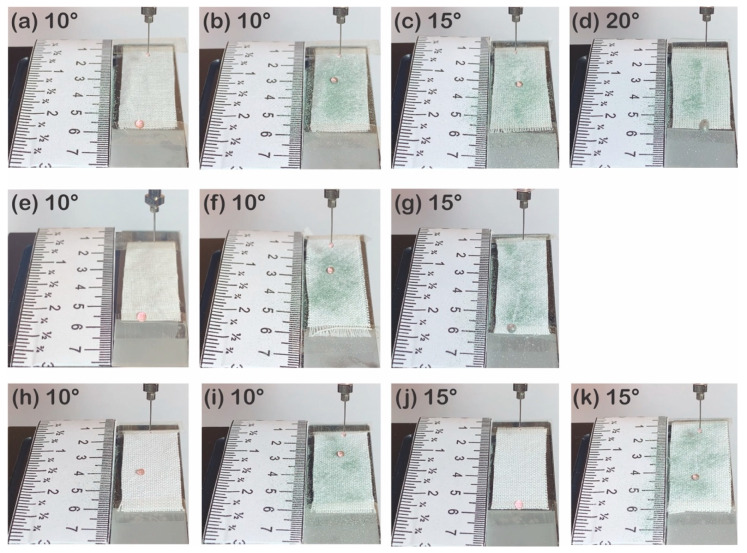
Self-cleaning test of P1T4 PDMS-coated 177D without silicon carbides at 10° (**a**) and with silicone carbides at 10° (**b**), 15° (**c**) and 20° (**d**), P1T4 PDMS-coated 314D without silicon carbides at 10° (**e**) and with silicon carbides at 10° (**f**), 15° (**g**) and P1T4 PDMS-coated 475D without silicon carbides at 10° (**h**) and 15° (**j**) and with silicone carbides at 10° (**i**) and 15° (**k**).

**Figure 14 polymers-13-00817-f014:**
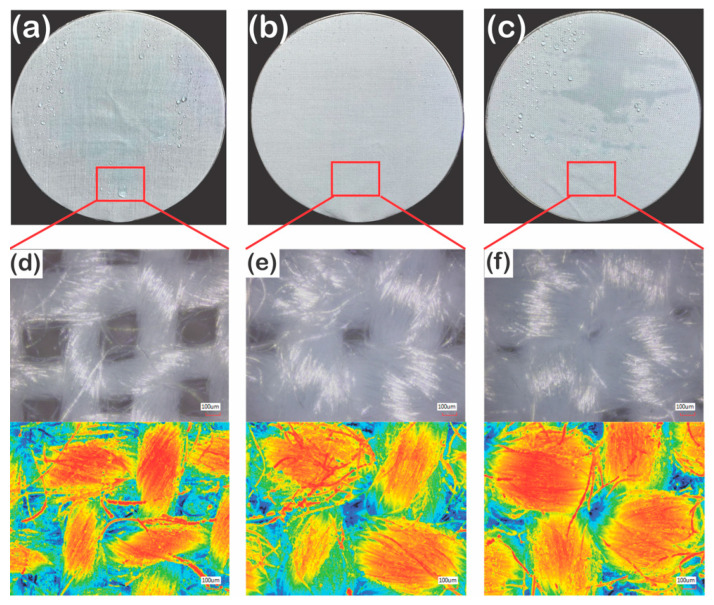
Photographs after water spraying test and optical images (**upper**) and laser images (**down**) of confocal microscope of 1% 4 min PDMS-coated PET fabrics: 177D (**a**,**d**), 314D (**b**,**e**), and 475D (**c**,**f**).

**Figure 15 polymers-13-00817-f015:**
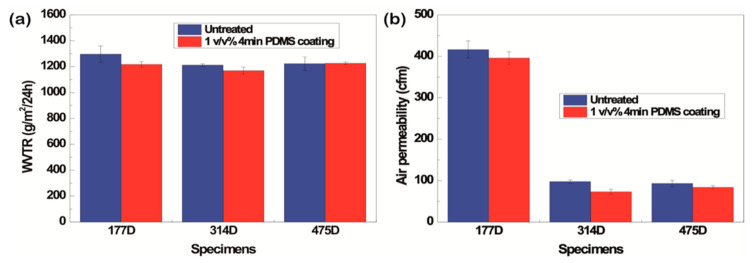
Water vapor transmission rate (**a**) and air permeability (**b**) before and after P1T4 PDMS coating.

**Table 1 polymers-13-00817-t001:** Characteristics of the polyester fabrics.

Fiber	Weave	Sample Code	Fiber Type	Yarn Count	Twist (turns/m)	Weave Density (Wrap/Weft)	Cover Factor		Cloth Cover Factor	Thickness (mm)	Solid Volume Fraction
							Wrap	Weft			
PET	Plain	Staple fabric	Staple	177D	865.8	66/66	12.1	12.1	19.0	0.28 ± 0.01	0.33
		Filament fabric	Filament	150D/144f	0	72/72	12.1	12.1	19.0	0.21 ± 0.01	0.38
		177D	Staple	177D	865.8	53/53	9.7	9.7	16.0	0.28 ± 0.01	0.24
		314D	Staple	314D	570.8	40/40	9.7	9.7	16.0	0.45 ± 0.01	0.33
		475D	Staple	475D	491.2	32/32	9.6	9.6	15.9	0.55 ± 0.01	0.33

**Table 2 polymers-13-00817-t002:** Sample codes depending on concentrations and durations of polydimethylsiloxane (PDMS) coating.

PDMS Coating		
Concentration (*v*/*v*%)	Duration (min)	Code
1	1	P1T1
	4	P1T4
	7	P1T7
4	1	P4T1
	4	P4T4
	7	P4T7
8	1	P8T1
	4	P8T4
	7	P8T7

**Table 3 polymers-13-00817-t003:** Static contact angles of 177D, 314D, and 475D according to PDMS concentrations (1, 4, and 8 *v*/*v*%) and coating durations (1, 4, and 7 min).

Static Contact Angle (°)									
	177D			314D			475D		
*v*/*v*% min	1	4	7	1	4	7	1	4	7
1	168.1 ± 3.0	168.3 ± 2.6	164.1 ± 2.1	0.0 ± 0.0	168.0 ± 1.6	169.0 ± 2.5	0.0 ± 0.0	165.7 ± 2.6	167.6 ± 2.4
4	168.6 ± 0.4	166.5 ± 2.1	164.8 ± 1.5	118.7±32.6	167.2 ± 2.1	168.8 ± 2.6	0.0 ± 0.0	165.7 ± 1.4	168.2 ± 2.0
8	166.1 ± 1.6	166.9 ± 1.3	163.5 ± 2.2	154.6±10.4	167.1 ± 2.7	168.9 ± 3.5	105.3 ± 48.7	166.7 ± 3.2	168.4 ± 1.2

**Table 4 polymers-13-00817-t004:** Shedding angles of 177D, 314D, and 475D according to PDMS concentrations (1, 4, and 8 *v*/*v*%) and coating durations (1, 4, and 7 min).

Shedding Angle (°)										
		177D			314D			475D		
	min	1	4	7	1	4	7	1	4	7
*v*/*v*%										
1		8.8 ± 0.8	8.5 ± 0.0	10.2 ± 1.8	45.0 ± 0.0	8.8 ± 0.3	9.5 ± 0.0	45.0 ± 0.0	12.5 ± 0.5	11.5 ± 0.3
4		8.3 ± 0.6	10.3 ± 0.3	10.5 ± 0.9	45.0 ± 0.0	9.7 ± 0.8	9.4 ± 0.3	45.0 ± 0.0	11.2 ± 0.3	11.7 ± 1.0
8		11.3 ± 1.2	12.7 ± 1.0	13.0 ± 1.3	17.3 ± 5.9	9.3 ± 1.5	9.4 ± 0.3	45.0 ± 0.0	11.8 ± 0.6	11.6 ± 0.6

## Data Availability

The data presented in this study are available in the Appendix A.

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
