# Peer review of "The Effect of Fiber Type and Yarn Diameter on Superhydrophobicity, Self-Cleaning Property, and Water Spray Resistance"

_polymers, 2021, doi:10.3390/polym13050817_

Round 1

Reviewer 1 Report

The authors have improved the article. I only suggest the equations should be typed via Mathtype software. After the modification, it can be published.

Author Response

The authors thank you for the suggestion. However, unfortunately, the math type cannot be downloaded now. The equation now the authors are using is the Latex equation, which is a math markup language familiar to many in science and math and used widely. Therefore, the authors think there are no problems with the current equations. But, the authors will try to use the math type when it is available. Thank you.

Reviewer 2 Report

The authors correctly replied to the previous requests and the paper has been improved, so in my opinion in this version the manuscript is suitable for publication

Author Response

Dear. reviewer,

We appreciate your confirmation about the reviews and revised version of the manuscripts.

Reviewer 3 Report

Title: The Effect of fiber type and yarn diameter on superhydrophobicity, self-cleaning property, and water spray resistance

The authors worked with the effects of fiber type and yarn diameter on superhydrophobicity, self-cleaning and water spray resistance properties. Recent work related to self-cleaning and superhydrophobicity should be cited in this manuscript. Just for the improvement of the manuscript, I recommend the authors to cite the following papers:

  1. One-pot sonochemical synthesis of ZnO nanoparticles for photocatalytic applications, modelling and optimization”. Materials, 13(1), (2020), 1-18, 14. https://doi.org/10.3390/ma13010014.

  1. Functional properties of sonochemically synthesized zinc oxide nanoparticles and cotton composites”. Nanomaterials, 10(9), (2020), 1-14, 1661. https://doi.org/10.3390/nano10091661.

  1. Enhanced mechanical properties of eucalyptus-basalt based hybrid reinforced cement composites”. Polymers, 12(12), (2020), 1-15, 2837. https://doi.org/10.3390/polym12122837.

Statistical analysis is required to judge the significance of the results for this work. I recommend the authors to perform ANOVA and Regression analysis and add the results in the manuscript.

Decision: Minor Revision

Author Response

Dear. reviewer,

The authors appreciate your comments and revised the manuscript as follows,

line 41,  page 1

Many techniques have been developed to render surface roughness [8–10]

Reviewer 4 Report

Superhydrophobicity has been extensively investigated not only for biomedical products but also for clothing textiles because superhydrophobic surfaces can make the particular cells attract or repel water droplets easily on their surfaces. In this paper, this study compared superhydrophobicity and water spray repellency based on the fiber types and yarn diameters. Regardless of the fiber types, filament fabric and staple fabric showed decreased static contact angles and increased shedding angles by increasing the PDMS coating concentrations and durations. Besides, the effect of fiber type and yarn diameter on superhydrophobicity and water spray resistance was studied. The results suggest that superhydrophobicity can be achieved with micro/micro hierarchical structures and the mono microstructure is favorable for the self-cleaning property and water spray repellency. Furthermore, superhydrophobic and water spray repellent fabric can be achieved by modifying the micro/micro hierarchical structures only by simple PDMS coating. The topic is important, the results are interesting and the methodology followed is appropriate, while the content falls well within the scope of this Journal. In general the paper makes fair impression and my recommendation is that it merits publication in this Journal, after the following major revision:

  1. The introduction should be reconstructed to present a coherent literature review. It may help the authors by answering the following questions: Why are these works relevant? Which specific problems were addressed? How are the previous results related with the latest work? What are the outstanding, unresolved, research issues? Answering the questions leads to the novelty of the proposed work naturally.
  2. The Knudsen number is an important parameter of micro-scale flow, the change of which will affect the micro-scale effect of gas mass transfer in porous fiber. Thus, the Knudsen number may have an important influence on air permeability.
  3. Experiment part. Although the results look “making sense”, the current form reads like a simple lab report. The authors should dig deeper in the results by presenting some in-depth discussion.
  4. The authors should give some detailed description of the experiments such as Materials, Hydrophobization, Characterization, etc.
  5. In Fig.5, the authors should give the explanations for the difference of data collected from different sources.
  6. When the yarn diameter increased, the fabrics needed higher PDMS concentrations and longer coating durations for uniform coating. The water vapor transmission rate and air permeability did not change significantly after coating. The authors should give some explanation on above results and analyze the physical mechanism in detail.
  7. Fiber reinforced polymer composite has been widely used in many fields of life. The surface roughness (see [A fractal model for capillary flow through a single tortuous capillary with roughened surfaces in fibrous porous media, Fractals, 2021, 28(4):2150017]), aspect ratio, diameter and flexibility of porous fibers (see [Powder Technology, 2019, 349:92-98; International Journal of Heat and Mass Transfer, 2019, 137:365-371]) are quite suitable for textiles. Authors should introduce some related knowledge to readers.
  8. There are also some grammar issues in the text. The authors are required to make a revision throughout the whole manuscript to improve the English writing thoroughly and carefully.

Round 2

Reviewer 4 Report

Basically, my comments have been addressed.

This manuscript is a resubmission of an earlier submission. The following is a list of the peer review reports and author responses from that submission.

Round 1

Reviewer 1 Report

In the current study, the authors have comprehensively investigated effects of fiber type and yarn diameter on superhydrophobicity, self-cleaning and water spray resistance. They got many useful conclusions on material components and morphological observations. The static contact angle and shedding angle were both measured. The findings may be useful for many areas, such as textile, material design and surface pattern.

In my viewpoint, the language sounds good, and the logic of the paper is well organized. I think it can be accepted for publication after some minor modifications.

  1. In many areas, the term “superhydrophobicity” and “self-cleaning” cannot be clearly distinguished. Herein, they represent different meanings, and the authors should mention this issue again.
  2. The plasma technique has already been applied to alter the wettability of solid surface, such as rock surface. Please see the recent work: Yu H., et al., Wettability enhancement of hydrophobic artificial sandstones by using the pulsed microwave plasma jet. Colloid and Interface Science Communications, 2020, 36: 100266.
  3. Is the meaning of “shedding angle” the same as the “sliding angle” in many references?
  4. (1) can be expressed as a formulas including the physical quantities.
  5. After each equation, there should be “,” or “.”.
  6. The authors used Wenzel model to illustrate the macroscopic angle, and this should be further considered. How about the Cassie model? Actually in many cases the Cassie-Baxter model better applies to the experimental result. See: Liu J. L. et al., Mechanisms of superhydrophobicity on a hydrophilic substrate. Journal of Physics: Condensed Matter, 2007, 19(35): 356002.
  7. In Eq. (2), R is not the radius of the “meniscus”. Please check it.
  8. In Eq. (2), the advancing angle should be properly cited.
  9. Most of the current work is experiment. For a detailed analysis, the modeling and simulation must be carried out. This can be discussed in the outlook of the future work.
  10. Normally, the “hierarchical” structure represents the “micro-nano composite” structure, spanning different scales. Herein, it has different meanings. Please make sure the actual definitions of these terminologies.
  11. For the water penetrating into pores, the classical wetting models (Wenzel and Cassie) do not apply. In this case the penetration equation must be considered.

Reviewer 2 Report

The aim of this paper should be the study of the wettability properties of PDMS coating on fibres, comparing the effect of different parameters. But, in my opinion, even the paper contains a lot of interesting experimental data, in this form seems more similar to an uncompleted scientific reports instead of a complete scientific work, lacking its focus. 

Just some observations

  • Introduction: line 36-37, please check that sentences, it is not clear the link between the specific technique (plasma etching) and the silk fibers. Maybe the sentence has to be reformulated.

Note also that there are even recent examples in which plasma etching and plasma deposition have been combined to obtain a superhydrophobic material (see for examples: “Ultra hydrophobic/superhydrophilic modified cotton textiles through functionalized Diamond-Like Carbon coatings for self-cleaning applications” D. Caschera, B. Cortese, A. Mezzi, M. Brucale, G. Ingo, G. Gigli, G. Padeletti Langmuir 29, (2013) 2775−2783; Morphological and Chemical Effects of Plasma Treatment with Oxygen (O2 ) and Sulfur Hexafluoride (SF6 ) on Cellulose Surface J. Sanches Gonzaga de Camargo, A. Junior de Menezefoss , N. Cristino da Cruz , E. Cipriano Rangel , A. de Oliveira Delgado-Silva Materials Research. 2017; 20(Suppl. 2): 842-850; Mahdieh Z.M., Shekarriz S., Taromi F.A. (2020) Fabrication of Super-Hydrophobic Fabric by Eco-Friendly Plasma Technique. In: Mirzadeh H., Katbab A. (eds) Eco-friendly and Smart Polymer Systems. ISPST 2018. Springer, Cham.)

  • XPS analysis is not reported, not even in the SI, on the contrary of what declared in the paper. Since XPS is a powerful method to assure the successful of the PDMS deposition and understand if and how the coating is strongly bonded on the subtrates, it is mandatory to add and discuss this experimental data.
  • Line 182-184: the authors said that “As shown in Figure 5a, static contact angles of the filament fabric decreased with the increase in the PDMS concentration and coating time, caused by covering the surface with the solution uniformly”, but the increase of covering of the surface, depending on PDMS deposition parameters is not shown…Please add SEM/XPS analysis supporting this concept. I expexted that increasing concentration and time deposition, also thickness and morphology could change, influencing the hydrophobicity of the system.
  • AFM measurements could also be useful to evaluate the different roughness values, since in the paper is often mentioned as an important parameter, influencing the final properties
  • Caption of Figure 4: to avoid confusion maybe it is better to group staple fabric untreated and P1T1 coated and then untreat and P1T1 coated fabrics…nevertheless, check the actual caption, maybe there is a mistake in indexing the right figures (c is lost…)
  • Figure 5 is quite strange…if the WCA were measured for 3x3 different parameters (3 different concentrations and 3 dfferent times) for 2 different substrates, we expected to see at least 18 measurements and not only 6…maybe figure 5, on the contrary of what declared in the caption, contains only measurements referred to 1x3 conditions and 2 substrates…. (Figure 5. Static contact angles (a) and shedding angles (b) of staple and filament fabrics after 1, 4, and 8 v/v% PDMS coating for 1, 4, and 7 min.) In this way, the effect of different parameters on the final properties is not clear…again.
  • Lines 222/224/226/230…maybe is figure 6 and not figure 7…
  • Since all the previuos tests have been carried out using P1T1, it is not clear why the authors decided to used different conditions for studying the influence of the yarn dimension of the wettability properties…why P1T4 should have considered now?
  • Moreover, the same problem about a complete structural characterization is present in the second part of the paper. XPS is lost and the EDS data have been presented only for one particular condition (P1T4), without explanation about this choice.
  • Since the auhtors decided to choose one reference (P1T1) in the first part, and another (P4T1) in the second, this does not permit to compare the obtained results, as authors said in the conclusion “This study compared superhydrophobicity and water spray repellency based on the fiber types 469 and yarn diameters.”. Moreover, the same authors in the conclusion summurized the results obtained but they are not compared….
  • Ref 2-10-16-17-18-29-46-57 are not complete

In my opinion, in this form, the paper is not suitable for publication

Reviewer 3 Report

Manuscript ID: polymers-1010010

Title: The Effect of fiber type and yarn diameter on superhydrophobicity, self-cleaning property, and water spray resistance

The authors worked with the effects of fiber type and yarn diameter on superhydrophobicity, self-cleaning and water spray resistance properties. The topic of this study is old. A lot of studies have already done with the effects of fiber type especially cotton and polyester in staple and filament form. Although the content of this manuscript is not suitable for polymers journal. Therefore, I reject this manuscript.

Just for the improvement of the manuscript, I recommend some suggestions:

  1. What’s the novelty of this study when compared to other relevant studies?
  2. Statistical analysis is required to judge the significance of the results for this work. I recommend the authors to perform ANOVA and Regression analysis and add the results in the manuscript.
  3. The introduction part is lacking with important and necessary information relevant to this topic.
  4. I recommend this manuscript in a textile relevant journals rather than polymers.

Decision: Reject